# A Notch Filter-Based Coupling Circuit for UNB and NB PLC Systems

**DOI:** 10.3390/s22249722

**Published:** 2022-12-12

**Authors:** Luís Guilherme da Silva Costa, Wesley Mateus Cantarino, Ândrei Camponogara, André Augusto Ferreira, Moisés Vidal Ribeiro

**Affiliations:** 1Department of Electrical Engineering, Federal University of Juiz de Fora, Juiz de Fora 36036-900, Brazil; 2Department of Electrical Engineering, Federal University of Paraná, Curitiba 80060-000, Brazil

**Keywords:** power line communication, coupling circuit, notch filter, narrowband, ultra-narrowband

## Abstract

This paper introduces an analog notch filtering-based coupling circuit for receivers in ultra-narrowband and narrowband power line communication systems, which are connected to low-voltage electric power grids. It is composed of a twin-T notch analog filter, which is responsible for imposing a significant attenuation on the main frequency (i.e., $f_{0}\in \left\{50,60\right\}$ Hz) in cascade with an elliptic low-pass analog filter, designed with a 3 dB cut-off frequency of $f_{c}\gg f_{0}$. For $f_{0}=60$ Hz and $f_{c}=2$ MHz, the prototype of the analog notch filtering-based coupling circuit attains attenuation values of 22 dB and less than 2 dB at the main frequency and in the rest of the frequency bandwidth, respectively, when practical scenarios are considered. Lastly, it shows that the analog notch filtering-based coupling circuit is more effective than a typical capacitive coupling circuit when frequencies lower than 3 kHz are considered for data communication and sensing purposes.

## 1. Introduction

The development of power line communication (PLC) systems dates back to the early 1900s when power line carrier technologies were used for telephony-based control over long distances [1]. Currently, PLC systems have been well investigated for indoor (in-home [2], in-building [3], and vehicle [4]) and outdoor (medium-voltage [5] and low-voltage [6,7]) electric power circuits. Electric power systems are data communication media for assisting low-cost and pervasive data networks for in-home and smart grid applications [8].

Conversely, electric power systems were not designed for data communication purposes, and, as a consequence, information-carrying signals transmitted through them may experience severe degradation [9]. The severe signal degradation is due to impedance mismatching, loads‘ dynamics and electromagnetic interference [10,11]. Note that electromagnetic interference results from the use of electromagnetically unshielded power cables. Additionally, the connection of PLC devices to power lines becomes a complex task when the amplitude of the mains signal is high, which is observed in medium voltage (MV) and high voltage (HV) electric power systems, or the transmission of PLC signals is close to the main’s frequency [12,13]. Consequently, PLC coupling devices require study [14].

There are studies covering coupling circuits for narrowband (NB) (10–500 kHz) and broadband (BB) (5–15 MHz) PLC systems [15,16]. However, to the best of our knowledge, there is a lack of studies on PLC coupling circuits for ultra-narrowband (UNB) PLC systems, which are characterized by meager data rates, the use of frequencies below 3 kHz, and distances up to 150 km [17]. UNB PLC systems are normally used for automatic meter reading (AMR) in rural areas that are difficult to access. There are UNB PLC systems on the market, such as Turtle System and two-way automatic communication system (TWACS) [18]. As a UNB PLC system is a simple and low-cost technology, it can also be applied to constitute low-power data networks for several Internet of Things (IoT) applications.

Coupling with electric power systems for data communication purposes in ultra-low frequencies is challenging because the proximity between the main’s frequency and the frequencies used by UNB PLC systems imposes constraints on the design of PLC coupling circuits. For instance, the attenuation of the main’s frequency must be high, while the attenuation of the other frequencies must be low. Currently, the design of PLC coupling circuits tries to accomplish this by combining a capacitor with an radio frequency (RF) transformer, which works as a high-pass analog filter in cascade with a low-pass analog filter. However, such a design may not be helpful for UNB PLC systems because a large-value capacitor is required. The use of such a large capacitor raises severe concerns in practical applications. Therefore, for advancing UNB PLC systems for new applications, it is necessary to research PLC coupling circuits that severely attenuate the main’s frequency and, at the same time, insert a low attenuation in the other frequencies. In other words, the coupling device must behave like an analog stop-band filter with a very narrow 3 dB bandwidth that, in the end, can be used for both UNB and NB PLC systems.

In this regard, this study focused on the design of the NB–PLC notch coupling circuit for receiving signals in low-voltage electric power grids. The proposed coupling circuit operates in the frequency band between 1 and $f_{c}$ Hz and therefore covers UNB and NB PLC systems. In this regard, the following contributions are presented:The proposition of a twin-T notch analog filter [19] based on an operational amplifier for imposing a high attenuation in the main’s frequency (e.g., 50 or 60 Hz) and a low attenuation in the frequencies different from the main’s frequency.The description of a twin-T notch analog filter cascaded with a fifth-order low-pass analog elliptic filter with a 3 dB cut-off frequency $f_{c}\gg f_{0}$, in which $f_{0}$ is the main’s frequency. The order and type of the analog low-pass filter fulfill the need for having low attenuation in the frequency between 1 and $f_{c}$ Hz and a short transition bandwidth.The combination of electric protection with isolation circuits at the input and output of the NB–PLC notch coupling circuit for ensuring electric safety conditions of PLC receivers.

Analyzing the design and prototype of the NB–PLC notch coupling circuit, the following findings deserve attention:The use of a twin-T notch analog filter allows the design of PLC coupling devices for receiving PLC signals that are transmitted in the low frequencies (i.e., |f|<3 kHz and in the vicinity of the mains frequency). Additinoally, using electric protection and isolation circuits between the electric power grid and the twin-T notch analog filter ensures the conditions for the safe operation of the twin-T notch analog filter based on an operational amplifier. Moreover, the electric protection, which is applied after the analog and elliptic low-pass filter, introduces additional electric safety conditions for PLC receivers.The simulation results of the NB–PLC notch coupling circuit show an attenuation of 57 dB in the main’s frequency. Additionally, a 3 dB frequency bandwidth equal to the frequency bandwidth is 246 Hz with lower- and upper-frequency edges of 11 Hz and 257 Hz, respectively. Additionally, an attenuation lower than 1 dB is achieved in most of the resting frequency bandwidth.The performance results of the prototype of the NB–PLC notch coupling circuit when $f_{c}=2$ MHz show an attenuation of 33 dB at the main’s frequency when impedance matching is ensured. Additionally, the results show a 3 dB cut-off when the frequency bandwidth is 246 Hz with lower- and upper-frequency edges equal to 11 Hz and 257 Hz, respectively, and an attenuation lower than 1 dB in most of the resting frequency bandwidth.The performance results obtained from the field trial show that the prototype of the NB–PLC notch coupling circuit introduces an attenuation of 22 dB to the main’s frequency and an attenuation lower than 1 dB after the transition frequency bandwidth. The reduction in the attenuation at the main’s frequency is caused by impedance mismatching with the electric power grids, characterized by dynamic access impedance.A field trial performance comparison between the prototypes of the proposed NB–PLC notch coupling circuit and a typical NB–PLC capacitive coupling circuit shows that the former is more effective than the latter when frequencies close to the main’s frequency are considered for data communication by PLC systems.

The rest of this paper is organized as follows: Section 2 provides a brief discussion on PLC coupling circuits. Section 3 details the proposed NB–PLC notch coupling circuit. Section 4 presents our simulation results, whereas Section 5 discusses the field trial results obtained by using the prototype of the NB–PLC notch coupling circuit. Finally, Section 6 provides concluding remarks.

## 2. Problem Formulation

The measurement and characterization of electric power grids for data communication purposes have been pursued for more than one century [20,21,22]. Currently, it has been well-established that NB PLC systems are suitable for outdoor scenarios (e.g., low data rates for smart grids [23,24]). In contrast, BB PLC systems are more advantageous in indoor scenarios (e.g., high-speed in-home data networks [25,26]). As electric power systems are dynamic and complex, they require research to advance PLC coupling circuits because these circuits significantly impact the performance of PLC systems.

There are several studies on coupling circuits for BB and NB PLC systems in the literature; see [14,27,28,29,30,31,32,33,34,35,36] and the references therein. In [27], the authors presented an analog Butterworth band-pass filter design covering the frequency band of 1–30 MHz. In [28], a BB PLC coupling circuit with a T configuration and a frequency band of 1.8–30 MHz was described. Costa et al. [29] discussed the design of capacitive PLC coupling circuits for NB, considering the frequency band 9–500 kHz, and BB for three distinct frequency bands: 1.7–50, 1.7–100, and 1.7–500 MHz.

The design of an NB–PLC coupling circuit to improve impedance matching by varying the winding ratio of an RF transformer was discussed in [30]. Additionally, Refs. [31,32] investigated transformerless coupling circuits because they simplify the design, are lower cost, and minimize power losses. A transformerless passive coupling circuit constitutes an alternative to transformer-based coupling devices; however, its use imposes a severe constraint. To characterize indoor electric power circuits, Gassara et al. [33] detailed two PLC coupling circuits that operate in the frequency band of 9–500 kHz: one for the transmitter and the other for the receiver. For coupling with MV and HV electric power systems, line trap circuits were exploited in [34,35]. Additionally, [14] surveyed coupling circuits for both NB and BB PLC systems. Finally, in [36], a PLC coupling circuit for the frequency band of 40–90 kHz was proposed to measure PLC channels in low-voltage and outdoor electric power grids.

Most coupling circuits for extracting PLC signals from low voltage (LV) electric power grids apply a capacitor to block the main’s signal. Then, they use an RF transformer for galvanic isolation and a low-pass analog filter for limiting the spectrum content of the received signal, which will be digitized. The alternating current (AC)-blocking capacitor aims to offer low impedance for high frequencies, which are used by PLC systems, and a high impedance for low frequencies, such as the main’s frequency (i.e., 50 or 60 Hz).

The value of the AC-blocking capacitor defines the lower edge of the frequency band used by the PLC coupling circuit because the series capacitor works as a high-pass filter. However, the use of a capacitor in series results in significant attenuation in ultra-low frequencies and, consequently, UNB PLC systems become unfeasible. To illustrate this situation, Figure 1 shows the attenuation of an AC-blocking capacitor when its value varies from 1 to 11.16 μF. The results in Figure 1 show that an AC-blocking capacitor, which aims to severely attenuate the amplitude of the main’s frequency, may not be appropriate for UNB PLC systems because it also severely attenuates the signals occupying the whole UNB spectrum.

The main’s frequency can also be attenuated using a band-pass analog filter; however, the complexity and cost of attaining good performance may not be attractive. For instance, it would demand a large-value inductor, which is costly and unfeasible.

Section 3 discusses a coupling circuit for receiving PLC signals for low-voltage electric power grids to overcome the above-mentioned limitation.

## 3. The Proposed NB-PLC Notch Coupling Circuit

The design of a practical coupling circuit for PLC systems operating in low-voltage electric power grids must address the following issues [37,38]:The use of electrical protection against transients originated from the electric power grid that may cause permanent damage to PLC devices.The appropriate design of analog filters for ensuring that the PLC system correctly operates in the specified frequency band and, consequently, introduces low insertion and return losses in the working frequency band.The significant attenuation of the main’s frequency (i.e., 50 or 60 Hz) for avoiding damage to the electronic circuit due to the presence of high-voltage components in electric signals.The capacity to handle the resulting attenuation from impedance mismatching between the PLC coupling circuit and electric power grids due to the dynamic nature of loads.

Among the above-mentioned issues, attenuation of the main’s frequency is a serious concern to ensure the electrical safety of the operation of PLC receivers. In practice, a capacitor followed by a transformer is used to attenuate the main’s frequency; however, it results in high attenuation in low frequencies; see details in Section 2.

In this regard, we introduce the NB–PLC notch coupling circuit. Its scheme is shown in Figure 2. The main difference between the NB–PLC notch coupling circuit and its predecessors is the use of a twin-T notch analog filter to filter out the main’s frequency and provide a low attenuation in the other frequencies, which includes frequencies close to the main’s frequency. The proposed NB–PLC notch coupling circuit imposes high attenuation in the main’s frequency and a low attenuation in other frequencies (e.g., f<3 kHz). Consequently, it can be helpful in both UNB and NB PLC systems over low-voltage electric power grids. Note that the NB–PLC notch coupling circuit was designed to operate in single-phase low-voltage electric power circuits and the frequency band of 1 Hz to $f_{c}$, in which $f_{c}\leq 2$ MHz (A higher upper-frequency edge can also be considered by redesigning the low-pass analog filter). The proposed NB–PLC notch coupling circuit was also designed for 127 V${}_{\mathrm{rms}}$ single-phase electric power circuits; however, it can be applied to a higher low-voltage value (i.e., 220 V_rms_) if a resistive divisor is considered.

Detailed descriptions of each part of the proposed NB–PLC notch coupling circuit are provided in the following subsections.

### 3.1. Electrical Protection

PLC coupling circuits are occasionally subjected to high-voltage and current transients that can damage PLC transceivers. Consequently, surge protective devices (SPDs) must be applied to guarantee the immunity of PLC transceivers against such transients. The presence of the SPDs limits the voltage level at the input of the coupling circuit by either blocking or shorting to ground any unwanted voltage level above a safe threshold. Additionally, the SPD must simultaneously provide surge protection in all ports of the PLC coupling circuit and low insertion loss in the whole operating frequency band. The SPD in the NB–PLC notch coupling circuit is implemented in two stages, which offers enough protection to the rest of the NB–PLC notch coupling circuit.

The first stage refers to the component $U_{R}$, which usually is a metal oxide varistor (MOV) or gas discharge tube (GDT) component. Note that the use of each one of them depends on the working frequency, response time, and voltage level. The first stage also covers the fusible resistor (FR) in series, see Figure 2. It refers to the circuitry directly connected to the outlet for blocking voltage surges from the electric power grid. In our study, $U_{R}$ is a MOV because its response time is in nanoseconds, which is faster than the GDT, and presents high resistance that falls to tens of milliohm if the voltage level at its input is above a certain threshold. The FR component offers low resistance, which is in milliohms, for providing a limitation on the inrush current and introducing low insertion loss. In the end, the combination of FR and MOV components impose a limit on the overcurrent and inrush currents.

The second stage implements Zener diodes $Z_{1}$ and $Z_{2}$ in Figure 2. They operate as clamping diodes, and, as a consequence, they guard the fragile input of the PLC receiver against transient voltages and offer a high level of surge dissipation. Therefore, their operating voltage level and frequency bandwidth must be carefully chosen.

### 3.2. Isolation

Figure 2 shows that capacitors $C_{1}$ and $C_{2}$ are used to isolate the NB–PLC notch coupling circuit from the main’s signal. Capacitor $C_{1}$ blocks the direct current (DC) component, which supplies the operational amplifier integrated circuit (IC), while the capacitor $C_{2}$ aims to avoid the ground loop. The values of capacitors $C_{1}$ and $C_{2}$ must be carefully chosen because they work as high-pass analog filters and, consequently, affect the magnitude response of the NB–PLC notch coupling circuit, mainly at low frequencies. In our design, the values of $C_{1}$ and $C_{2}$ were obtained using advanced design system (ADS) software with the Filter Design Guide tool [39].

### 3.3. Twin-T Notch Analog Filter

The parallel implementation of low-pass and high-pass analog filters with a voltage follower amplifier results in a stop-band analog filter, the twin-T notch analog filter, which received its name owing to the T-shaped topology of the circuit centered at the nodes labeled *a* and *b* [40]; see Figure 2. According to this figure, resistors $R_{1}$ and $R_{2}$ and capacitor $C_{3}$ constitute a low-pass analog filter, while capacitors $C_{4}$ and $C_{5}$ and resistor $R_{3}$ constitute a high-pass analog filter. The twin-T notch analog filter aims to introduce a severe attenuation in the main’s frequency (e.g., over 40 dB) and low attenuation in the frequency of the main’s frequency.

To design the twin-T notch analog filter, we assume that $R_{1}=R_{2}=R$, $C_{4}=C_{5}=C$, $R_{3}=R/2$, and $C_{3}=2C$. Consequently, the sums of the currents in nodes *a* and *b*, shown in Figure 2, in the Laplace domain, are expressed as
(1)$$\frac{2V_{a}\left(s\right)-V_{i}\left(s\right)}{R}+2V_{a}\left(s\right)sC+\frac{V_{a}\left(s\right)-V_{o}\left(s\right)}{R}=0,$$
which results in
(2)$$V_{a}\left(s\right)[2+2sCR]-V_{o}\left(s\right)=V_{i}\left(s\right),$$
and
(3)$$\left[V_{b}\left(s\right)-V_{i}\left(s\right)\right]sC+\frac{2V_{b}\left(s\right)}{R}+[V_{b}\left(s\right)-V_{o}\left(s\right)]sC=0,$$
which results in
(4)$$V_{b}\left(s\right)\left[2sCR+2\right]-V_{o}\left(s\right)sCR=V_{i}\left(s\right)sCR,$$
in which $V_{a}\left(s\right)=L\left\{v_{a}\left(t\right)\right\}$ and $V_{b}\left(s\right)=L\left\{v_{b}\left(t\right)\right\}$ are the voltages at nodes *a* and *b*, respectively. Additionally, $V_{o}\left(s\right)=L\left\{v_{o}\left(t\right)\right\}$, $V_{+}\left(s\right)=L\left\{v_{+}\left(t\right)\right\}$, $V_{o}\left(s\right)=V_{+}\left(s\right)$, and L{·} refer to the Laplace transform operator. The high impedance seen by the inputs of the voltage-follower amplifier IC means that the differential voltage between the inverting ($v_{-}\left(t\right)$) and noninverting voltage inputs ($v_{+}\left(t\right)$) of the IC is assumed to be zero, i.e., $v_{+}\left(t\right)=v_{-}\left(t\right)$. Additionally, the IC voltage output is directly connected to its inverting voltage input forcing $v_{-}\left(t\right)=v_{o}\left(t\right)$ and, consequently, $v_{o}\left(t\right)=v_{+}\left(t\right)$. Such a circuit is a voltage-follower with a gain equal to 1. Note that it is an ideal circuit to be used as a constant voltage source or a voltage regulator because of the isolation propriety between its input and output. Consequently, it exploits the main advantage of the unity gain voltage follower configuration, which makes sense if the impedance matching or circuit isolation is more important than the voltage or current amplification.

Next, summing the currents away from the noninverting input terminal of the IC, we have
(5)$$\frac{V_{o}\left(s\right)-V_{a}\left(s\right)}{R}+[V_{o}\left(s\right)-V_{b}\left(s\right)]Cs=0,$$
which results in
(6)$$V_{o}\left(s\right)\left[1+sCR\right]-V_{a}\left(s\right)-V_{b}\left(s\right)sCR=0.$$

Now, we obtain $V_{o}\left(s\right)$ by solving the system of linear equations composed of (Equation 2), (Equation 4) and (Equation 6) through Cramer’s rule as follows: (7)$$\begin{array}{cc} V_{o}\left(s\right) & =\frac{\det\left(\left[\begin{array}{ccc} 2sCR+2 & 0 & V_{i}\left(s\right)\\ 0 & 2sCR+2 & V_{i}\left(s\right)sCR\\ -1 & -sCR & 0 \end{array}\right]\right)}{\det\left(\left[\begin{array}{ccc} 2sCR+2 & 0 & -1\\ 0 & 2sCR+2 & -sCR\\ -1 & -sCR & sCR+1 \end{array}\right]\right)}\\ \; & =\frac{V_{i}\left(s\right)\left(s^{2}C^{2}R^{2}+1\right)}{s^{2}C^{2}R^{2}+4sCR+1}, \end{array}$$
where det(·) denotes the determinant operator. As a result, the transfer function of the twin-T notch analog filter is expressed as
(8)$$\begin{array}{c} T\left(s\right)=\frac{V_{o}\left(s\right)}{V_{i}\left(s\right)}=\frac{s^{2}+1/{\left(CR\right)}^{2}}{s^{2}+4s/\left(CR\right)+1/{\left(CR\right)}^{2}}, \end{array}$$
which is the same expression of the transfer function of a stop-band analog filter. According to [41], the transfer function of a stop-band analog filter is given by
(9)$$\begin{array}{c} H\left(s\right)=\frac{K[{\left(s/\Omega_{0}\right)}^{2}+1]}{{\left(s/\Omega_{0}\right)}^{2}+2\zeta \left(s/\Omega_{0}\right)+1}, \end{array}$$
where $\Omega_{0}=2\pi f_{0}$ and $f_{0}$ is the notch frequency (e.g., 50 or 60 Hz), and ζ controls the gain of the filter frequency response. Comparing (Equation 8) and (Equation 9), we see that
(10)$$\begin{array}{c} R=\frac{1}{\Omega_{0}C}. \end{array}$$

As a standard procedure, we define $\Omega_{0}$, and then the *R* or *C* value is arbitrarily selected. However, due to the reduced set of values available in the market, the *C* value is usually chosen first, and then *R* is obtained from (Equation 10).

### 3.4. Elliptic Low-Pass Analog Filter

A low-pass analog filter is mandatory for correctly performing the analog-to-digital conversion process. In this sense, we designed a 5th-order elliptic low-pass analog filter, shown in Figure 2. The 5thorder elliptic low-pass analog filter is implemented by the capacitors $C_{6}$ to $C_{10}$ and inductors $L_{1}$ and $L_{2}$.

The elliptic structure was chosen because it has a steep roll-off, which results in better frequency selectivity than that of other analog filters, such as Butterworth and Chebyshev [42]. As the order of an elliptic analog filter is small, a low-cost implementation using a few components is attained. The cut-off frequency and the filter order are obtained according to the desired frequency selectivity [43]. A 5th-order elliptic low-pass analog filter with a cut-off frequency $f_{c}$ was designed using ADS software with the Filter Design Guide tool [39]. Table 1 summarizes the chosen specifications for designing this analog filter.

## 4. Simulation Results

This section discusses the numerical results we obtained using ADS software for simulating the proposed NB–PLC notch coupling circuit, which we designed to operate in frequency bands between 1 Hz and $f_{c}=2$ MHz at a notch frequency of 60 Hz. The choice of $f_{c}=2$ MHz as the upper-frequency edge shows that the proposed NB–PLC notch coupling circuit is adequate for UNB and NB PLC systems and energy harvesting in frequencies up to $f_{c}=2$ MHz, as investigated in [12].

Table 2 shows the components of the proposed NB–PLC notch coupling circuit when the main’s frequency is 50 and 60 Hz. Note that the operational amplifier, denoted by IC, is a TL-081 [44], with parameters listed in Table 3 [45]. Additionally, the values of the passive components were optimized to result in values available on the market. Furthermore, the output voltage of the practical operational amplifier is divided between this internal impedance and any external load impedance. The manufacturer’s specification data sheet is unclear if the impedance value refers to open-loop output impedance. In most practical applications, the operational amplifier is provided with feedback (i.e., closed-loop). In the data sheet of the TL-081, this impedance is specified (i.e., 125Ω). Consequently, we define an output impedance of 1 MΩ for the NB–PLC notch coupling circuit. It matches the input impedance of the oscilloscope used for characterizing the NB–PLC notch coupling circuit prototype; details are provided in Section 5.

In the following subsections, different simulations are presented. Each of them covers a relevant aspect of the prosed system.

### 4.1. Simulations #1

The notch frequency of the twin-T notch analog filter is also sensitive to nominal values of the passive components (i.e., resistors and capacitors) used to implement it. These passive components have a maximum deviation in their nominal values (tolerance range) ranging from 0 to 20%. In this sense, it is valuable to evaluate how the tolerance range impacts the notch frequency and, consequently, the magnitude of the frequency response of the NB–PLC notch coupling circuit.

Figure 3 shows the attenuation of the NB–PLC notch coupling circuit when a deviation up to ±20 % is applied to the nominal values of components *R* and *C* only because $R_{1}=R_{2}=R$, $R_{3}=R/2$, $C_{3}=2C$, and $C_{4}=C_{5}=C$. We assumed that *R* and *C* were uniform-distribution random variables centered in their nominal values to obtain these results. Note that we are only showing the attenuation values for the frequency band between 0 and 300 Hz to facilitate the visualization of detrimental effects of the variation in the components’ values on the performance of the twin-T notch analog filter. For example, Figure 3 also shows that the notch frequency may assume a value between 51 and 70 Hz when the attenuation value is between 26 and 58 dB. In other words, the tolerance of the components must be carefully chosen because it can significantly compromise the performance of the twin-T notch analog filter and, consequently, the proposed NB–PLC notch coupling circuit.

### 4.2. Simulations #2

Figure 4 shows the attenuation of the proposed NB–PLC notch coupling circuit, i.e., $20\log_{10}\left(\right|V_{\mathrm{out}}\left(f\right)/V_{\mathrm{in}}\left(f\right)\left|\right)$ dB, which we obtained by performing numerical simulations. An impedance value of 1 MΩ was used in both the input and output of the NB–PLC notch coupling circuit to obtain this result. The attenuation is flat and equal to 0 dB in the frequency range of 1 to 10 Hz. Additionally, the attenuation gradually increases to 57 dB at the frequency of 60 Hz. After that, the attenuation gradually grows to around ≈0 dB at a frequency of 600 Hz. Moreover, a relatively sharp roll-off can be observed, slightly faster near the 3 dB cut-off frequency of $f_{c}=2$ MHz. Additionally, we see a ripple between 0.9 and −2.01 dB in the operating frequency band. Moreover, the 3 dB cut-off frequency bandwidth is 246 Hz with lower- and upper-frequency edges equal to 11 Hz and 257 Hz. Overall, the curve plotted in Figure 4 shows that the proposed NB–PLC notch coupling circuit is suitable for both UNB and NB PLC systems.

### 4.3. Simulations #3

In this subsection, we compare the performance between the proposed NB–PLC notch coupling circuit with that of the NB–PLC capacitive coupling circuit, shown in Figure 5. The latter has $f_{c}=2$ MHz and introduces an attenuation equal to 50 dB at the main’s frequency. For more details, see [29]. As we can see, the NB PLC coupling circuit is composed of four building blocks. The blocks are the electric protection, high-pass analog filter ($C_{\mathrm{BLOQ}}$ capacitor), galvanic isolation (RF transformer with a 1:1 windings ratio), and analog low-pass filter (fifth-order low-pass elliptic analog filter). The cascade of a high-pass and a low-pass filter results in a band-pass filter. Note that the NB–PLC capacitive coupling circuit in [29] relies on a blocking capacitor to mitigate the presence of the main’s frequency. In other words, the capacitor $C_{\mathrm{BLOQ}}$ imposes a very high impedance to the main’s frequency (i.e., 50 or 60 Hz) and a very low impedance to the high-frequency spectrum, allowing high-frequency signals to pass through it.

Considering frequencies up to 9 kHz, Figure 6 shows the attenuation of the NB–PLC notch and NB–PLC capacitive coupling circuits. The NB-PLC notch coupling circuit shows an attenuation of 57 dB at a frequency of 60 Hz and 0 dB after 400 Hz. In contrast, the attenuation in the NB–PLC capacitive coupling circuit decreases with the increase in the frequency value (e.g., 78 dB in 0 Hz and 18 dB in 1 kHz) and attains an attenuation of 44 dB at 60 Hz. Overall, we can see that the NB–PLC notch coupling circuit can operate in the UNB spectrum, while the NB–PLC capacitive coupling circuit shows severe limitations when operating in the UNB spectrum.

## 5. Experimental Results

This section discusses the experimental results of the prototype of the proposed NB–PLC notch coupling circuit in the frequency band between 1 Hz and $f_{c}=2$ MHz. The prototypewasis mounted on a printed circuit board (PCB) using the components with their values listed in Table 2 and the operational amplifier model TL-081. As shown in Figure 7, PCB copper traces, components, a through-hole, and a surface mount device were mounted on both top and bottom layers of a doubled face PCB FR4 fiberglass board.

For field tests, we used an arbitrary function generator FY8300S [46] and an oscilloscope RTE 1204 Rhode & Schwarz [47] with input impedance set to 1 MΩ. In the first field test, impedance matching was ensured at both input and output of the NB–PLC notch coupling circuit (Section 5.1). The second field test addresses the situation in which the NB–PLC notch coupling circuit was connected to the electric power grid, meaning the impedance matching was not ensured (Section 5.2).

### 5.1. Experiment #1

This analysis was carried out to characterize the prototype’s performance when signal generator equipment was connected to its input while an oscilloscope was connected to its output. In both connections, impedance matching was ensured. The measurement setup, built to carry out this experiment, is shown in Figure 8.

To perform this experiment, the arbitrary function generator injected a voltage signal $v_{i}\left(f,t\right)=A_{i}\sin\left(2\pi ft+\theta \right)$ into the input of the prototype so that a voltage signal $v_{o}\left(f,t\right)=A_{o}\left(f\right)\sin\left(2\pi ft+\theta_{0}\right)+w\left(t\right)$ was collected from its output by the oscilloscope, where w(t) is the additive noise and f∈[0,2] MHz denotes the frequency range of values used by the arbitrary function generator. Note that $|S_{21}\left(f,t\right)|=|A_{o}\left(f\right)|/|A_{i}\left(f\right))|,\;\forall \;f$, and, as a consequence, the attenuation is given by $20\log_{10}\left(\right|S_{21}\left(f,t\right)\left|\right)$.

Figure 9 shows the attenuation of the NB–PLC notch coupling circuit using the measurement setup. We see that the twin-T notch analog filter gradually attenuated the voltage signal injected into the input of the NB–PLC notch coupling circuit from ≈0 dB at a 1 Hz frequency until it reached an attenuation of ≈33 dB at a frequency of 60 Hz, which is ≈24 dB lower than the attenuation obtained when impedance matching was ensured, see Figure 4. Additionally, its frequency bandwidth was 236 Hz with lower-frequency and upper-frequency edges equal to 14 Hz and 250 Hz, respectively, for the 3 dB cut-off frequency criterion. Overall, a significant attenuation of the mains signal was observed in practical scenarios.

### 5.2. Experiment #2

In this experiment, we used the measurement setup illustrated in Figure 10. Note that the input of the prototype was connected to the single-phase low-voltage of the electric power grid and an oscilloscope’s probe. In contrast, the output was connected to another oscilloscope’s probe. The oscilloscope’s probes connected to the input and output of the prototype allowed us to observe the performance of the proposed NB–PLC notch coupling circuit.

Measuring the voltage signal at the input of the proposed NB–PLC notch coupling circuit, Figure 11 shows that its value was 129.55 $V_{\mathrm{rms}}$ and the main’s frequency was ≈60.03 Hz. Now, measuring the output of the twin-T notch analog filter, we note that the measured voltage signal attained an amplitude of 10.41 $V_{\mathrm{rms}}$ and a main’s frequency of ≈60.01 Hz. Information about port Ch1 is provided in Figure 12. The measured voltage signal at the output of the NB–PLC notch coupling circuit showed an amplitude of 10.41 $V_{\mathrm{rms}}$ and a main’s frequency of ≈60.01 Hz; information about port Ch4 is provided in Figure 12. As $20\log_{10}\left(129.55/10.41\right)\approx 22$ dB, we see that an attenuation of 22 dB was imposed in the amplitude of the main;s frequency. Additionally, the root mean square (RMS) values at the output of the twin-T notch analog filter and the NB–PLC notch coupling circuit were equal, which means that the elliptic low-pass analog filter and electric protection stages introduced irrelevant performance losses.

Moreover, a comparison between the attenuation of the mains frequency, shown in Figure 9, and the reported results in the previous paragraph showed a performance loss of ≈11 dB. The lack of impedance matching between the electric power grid and the prototype was the source of this performance loss. Unfortunately, this problem is difficult to handle because the access impedance of electric power grids varies in time, space, and frequency [37]. An interesting direction to minimize this problem is to consider adaptive PLC coupling circuits, as described in [48].

Figure 13 shows the magnitude spectrum between 0 Hz and 1 kHz of the voltage signal measured at the input of the NB–PLC notch coupling circuit. Note that the power of the mains frequency is equal to ≈55 dBm. On the other hand, Figure 14 shows the magnitude spectrum between 0 Hz and 1 kHz of the voltage signal measured at the output of the NB–PLC notch coupling circuit. For this voltage signal, the power of the mains frequency is equal to ≈33 dBm. It means that the NB–PLC notch coupling circuit introduced an attenuation of ≈22 dB, which agrees with the attenuation calculated in the previous paragraph. Table 4 summarizes the powers of odd harmonics at the input and output of the NB–PLC notch coupling circuit. The results listed in this table show how the NB–PLC notch coupling circuit behaves in the UNB spectrum, in which harmonics are prominent.

## 6. Conclusions

This paper proposed the NB–PLC notch coupling circuit for coupling PLC receivers to LV electric power grids when UNB and NB spectra are considered for data communication purposes.

The simulation results showed that the combination of a twin-T notch analog filter and a fifth-order electric low-pass analog filter results in a PLC coupling circuit that is effective in both UNB and NB spectra. Additionally, the simulation results showed the advantage offered by the NB–PLC notch coupling circuit in comparison with the NB–PLC capacitive coupling circuit when the UNB spectrum is considered.

Moreover, the experimental results showed that ensuring impedance matching with the input and output of the proposed NB–PLC notch coupling circuit provides remarkable attenuation of the main’s frequency. Additionally, if the impedance matching is not ensured, representing practical scenarios, the main’s frequency is less attenuated; however, such attenuation is ≈22 dB, which is enough to reduce the amplitude of the voltage signal and perform the digitization process on good terms.

A further improvement related to the proposed NB–PLC notch coupling is the dynamic adjustment of the notch frequency and the dynamic impedance matching methods.

## Figures and Tables

**Figure 1 sensors-22-09722-f001:**
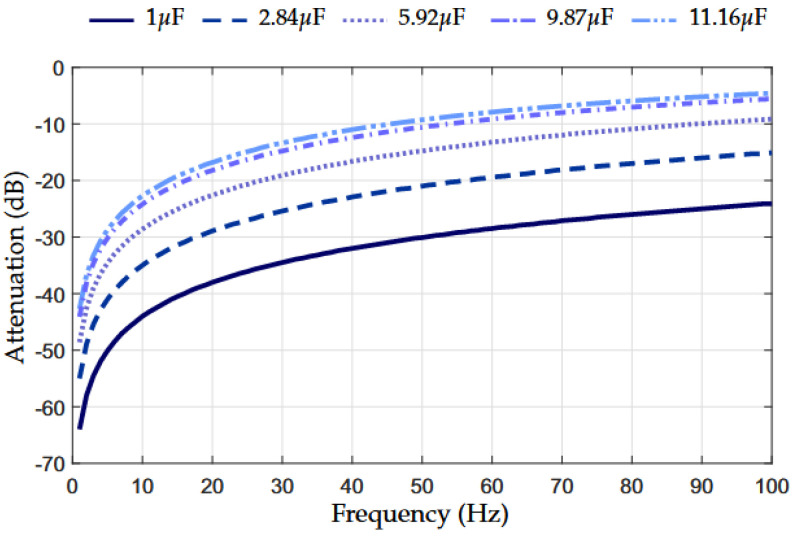
Attenuation profile of AC-blocking capacitor.

**Figure 2 sensors-22-09722-f002:**
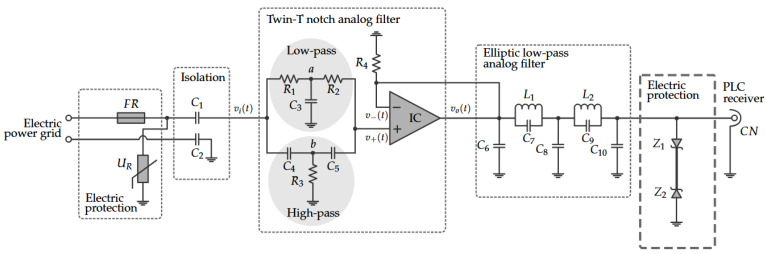
The scheme of the proposed NB–PLC notch coupling circuit.

**Figure 3 sensors-22-09722-f003:**
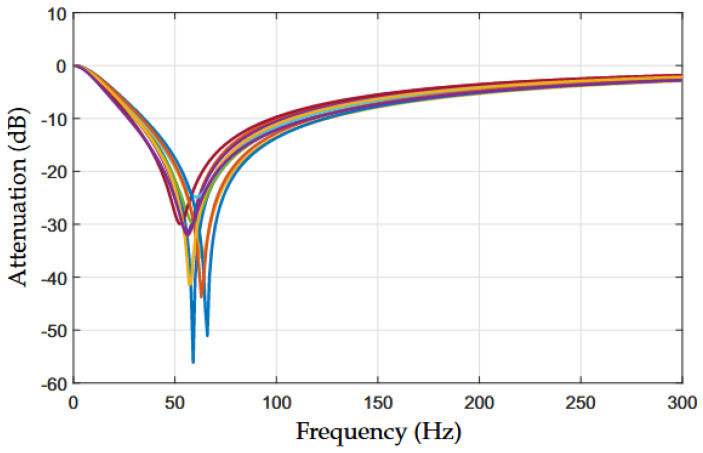
Numerical simulation of the attenuation of the twin-T notch analog filter for distinct deviations (up to ±20 %) from the nominal values of components $R_{1}$, $R_{2}$, $R_{3}$, $C_{3}$, $C_{4}$, and $C_{5}$.

**Figure 4 sensors-22-09722-f004:**
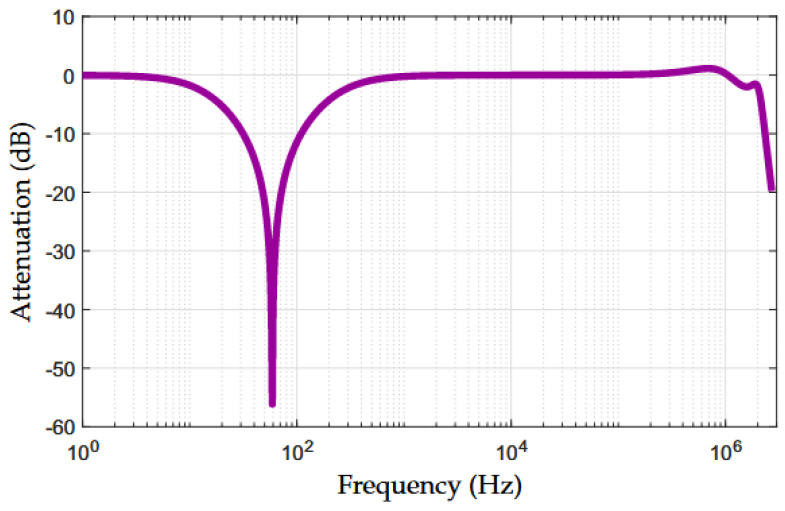
Attenuation of the simulated NB–PLC notch coupling circuit.

**Figure 5 sensors-22-09722-f005:**
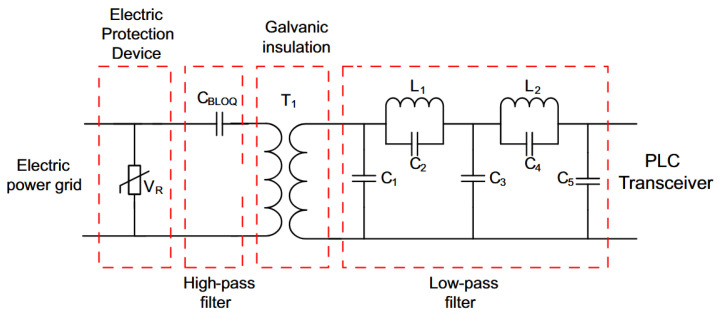
The block diagram of NB, capacitive, single-input single-output (SISO), and low-voltage PLC coupling circuit.

**Figure 6 sensors-22-09722-f006:**
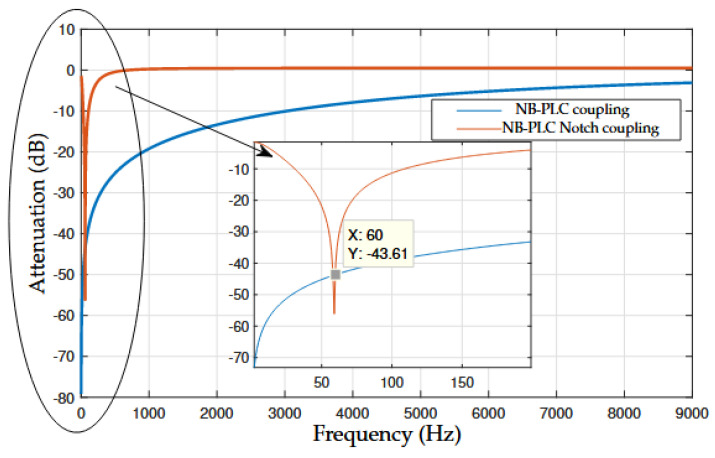
Attenuation comparison between NB–PLC notch coupling circuit and NB–PLC capacitive coupling circuit.

**Figure 7 sensors-22-09722-f007:**
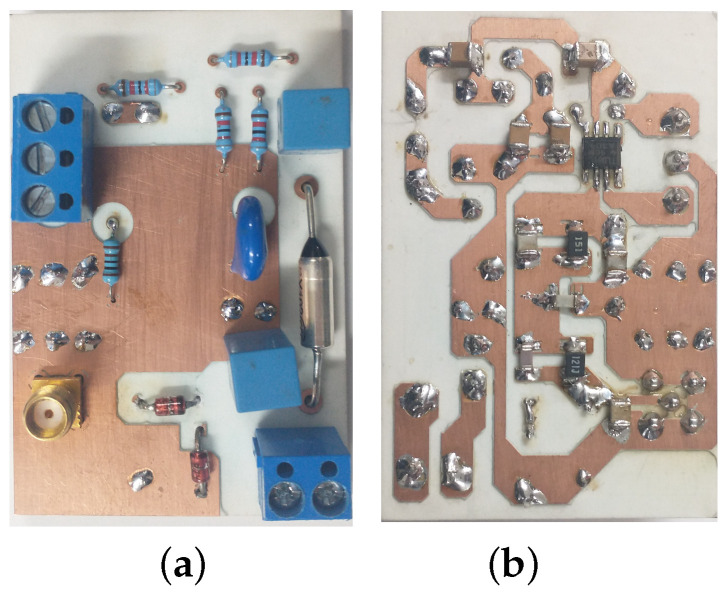
The prototype of the NB–PLC notch coupling circuit. (**a**) PCB top layer. (**b**) PCB bottom layer.

**Figure 8 sensors-22-09722-f008:**
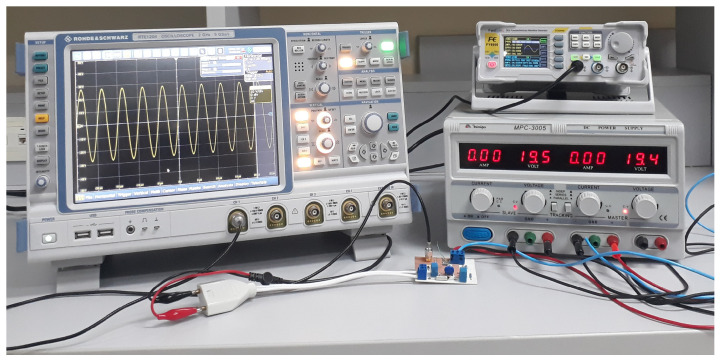
Measurement setup used to carry out Experiment #1.

**Figure 9 sensors-22-09722-f009:**
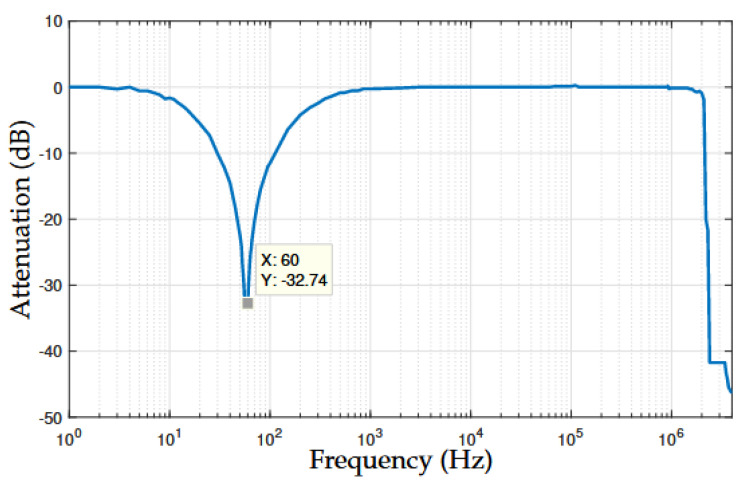
Attenuation of the prototype of the NB–PLC notch coupling circuit.

**Figure 10 sensors-22-09722-f010:**
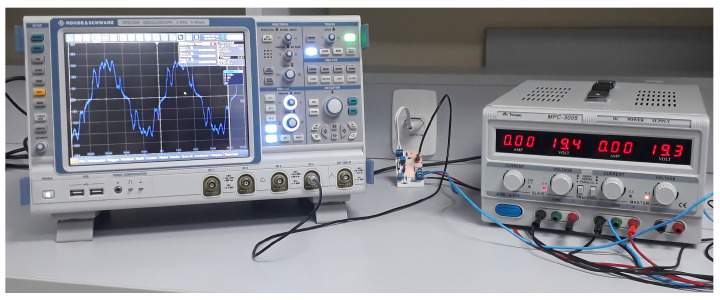
Measurement setup used to carry out Experiment #2.

**Figure 11 sensors-22-09722-f011:**
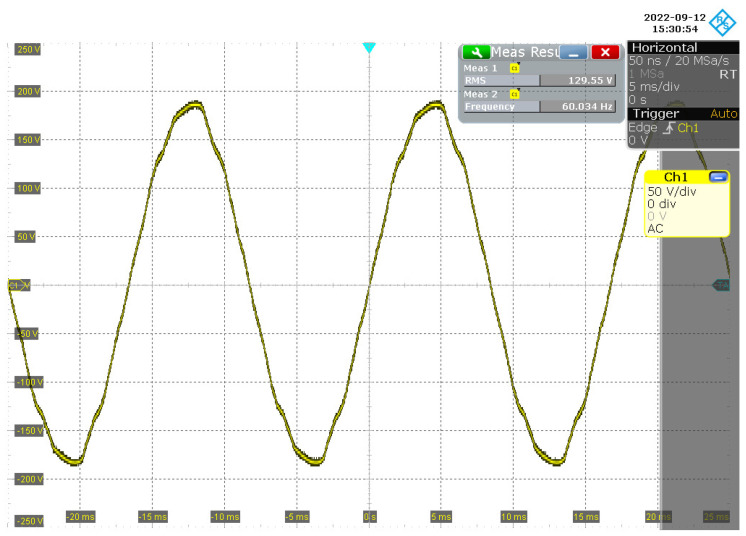
The waveform of the voltage signal at the input of the NB–PLC notch coupling circuit.

**Figure 12 sensors-22-09722-f012:**
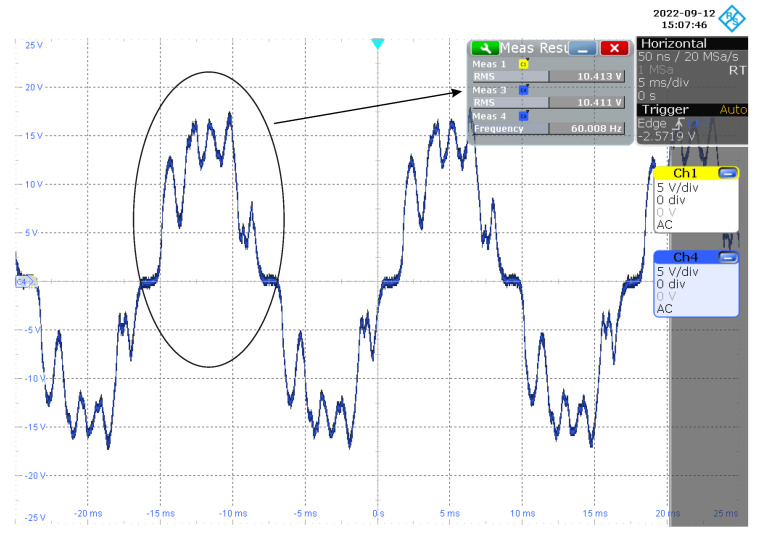
The waveform of the voltage signal at the output of the NB–PLC notch coupling circuit.

**Figure 13 sensors-22-09722-f013:**
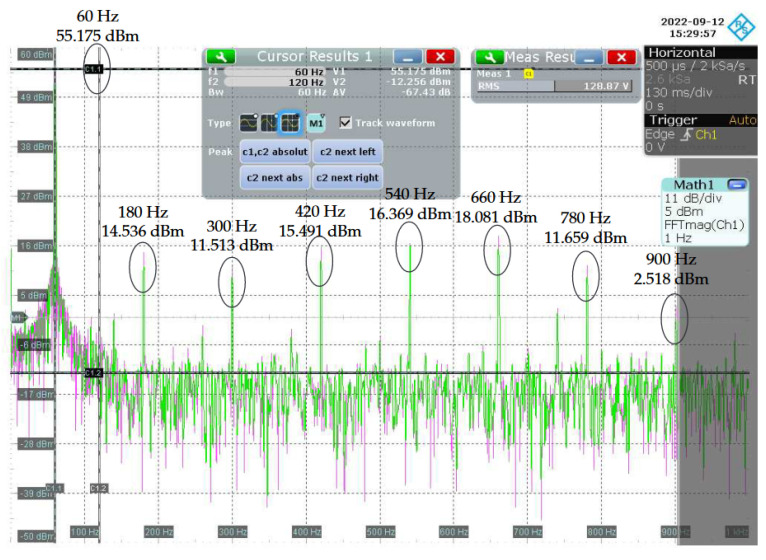
Magnitude spectrum of the voltage signal at the input of the NB-PLC notch coupling circuit.

**Figure 14 sensors-22-09722-f014:**
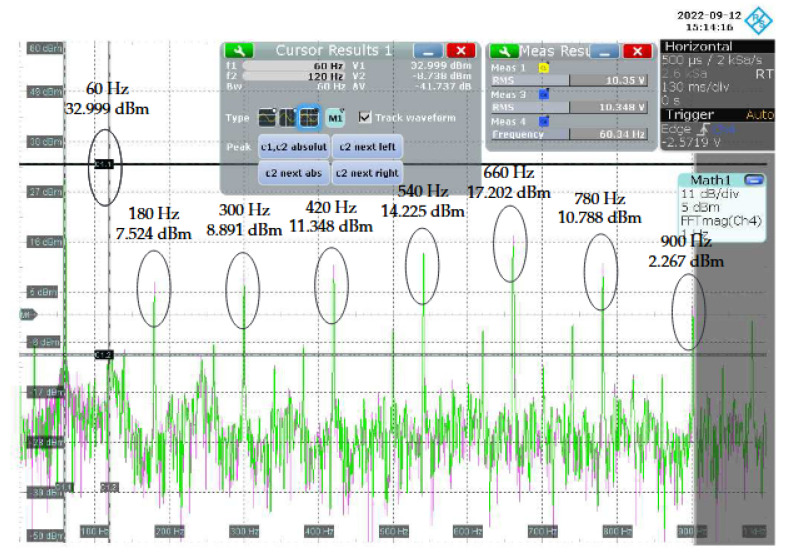
Magnitude spectrum of the voltage signal at the output of the NB–PLC notch coupling circuit.

**Table 1 sensors-22-09722-t001:** Specifications of elliptic low-pass analog filter.

Description	Value
Input impedance	2 kΩ
Output impedance	2 kΩ
Pass-band ripple	0.1 dB
Stop-band attenuation	40 dB

**Table 2 sensors-22-09722-t002:** The components of the NB–PLC notch coupling circuit prototype.

Components	Value at 50 Hz	Value at 60 Hz
$U_{R}$ (MOV)	250 V${}_{\mathrm{rms}}$	250 V${}_{\mathrm{rms}}$
FR	250 V${}_{\mathrm{rms}}$	250 V${}_{\mathrm{rms}}$
$Z_{1}$, $Z_{2}$	12 V	12 V
IC	TL081	TL081
$C_{1}$, $C_{2}$	2200 μF	2200 μF
$C_{3}$	66 nF	66 nF
$C_{4}$, $C_{5}$	33 nF	33 nF
$C_{6}$, $C_{10}$	1600 pF	1600 pF
$C_{7}$	180 pF	180 pF
$C_{8}$	2700 pF	2700 pF
$C_{9}$	470 pF	470 pF
$R_{1}$, $R_{2}$	96.5 kΩ	82 kΩ
$R_{3}$	48.25 kΩ	41 kΩ
$R_{4}$	1 kΩ	1 kΩ
$L_{1}$	4.7μH	4.7μH
$L_{2}$	3.9μH	3.9μH
$C_{N}$	SMA 50 Ω	SMA 50 Ω

**Table 3 sensors-22-09722-t003:** Parameter values of the operational amplifier model TL-081.

Parameter	Value
Gain	120 dB
Common mode rejection ratio	86 dB
Differential input resistance	25 kΩ
Differential input capacitance	3 pF
Common mode input resistance	90 MΩ
Common mode input capacitance	1 pF
Slew rate	13 V/μs
Unit gain bandwidth	3 MHz
Short-circuit output current	60 mA
Maximum symmetrical supply voltage	±18 V

**Table 4 sensors-22-09722-t004:** The power of the main and of a few harmonics components at the input and output of the NB–PLC notch coupling circuit.

Harmonic (Hz)	Input Power (dBm)	Output Power (dBm)
60	55.175	32.999
180	14.536	7.524
300	11.513	8.891
420	15.491	11.348
540	16.369	14.225
660	18.081	17.202
780	11.659	10.788
900	2.518	2.267

## Data Availability

Not applicable.

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
