# Peer review of "A Notch Filter-Based Coupling Circuit for UNB and NB PLC Systems"

_sensors, 2022, doi:10.3390/s22249722_

Round 1
Reviewer 1 Report
Please, find below some of my concerns;
1. Please avoid the lumped references in the introduction section.
2. The references are old. There are many recent papers related to the paper topic that must be added in the introduction section.
3. The paper needs English improvements. Proofreading by a native English speaker should be conducted to improve both language and organization quality
4. What are the research gaps in the literature? The authors did not identify the research gaps. The authors need to identify the research gaps, which they are planning to address in the paper.
5. The contributions of the paper must be better described. They must be benchmarked against the state-of-the-art to prove their novelty.
6. The conclusion should be revised. The reviewer strongly suggests improving the flow of the conclusion section. Start with a brief explanation of the paper, followed by a description of the main findings. Then finalize the conclusion with one or two suggestions for future work.
7. The conclusion needs to be more efficient and contains the paper outputs and outcomes.
Author Response
RESPONSE TO REVIEWER FILE
Industrial Sensors
TITLE: A Notch Filter-based Coupling Circuit for Ultra Narrowband and Narrowband PLC Applications
MANUSCRIPT ID: SENSORS-1969792
AUTHORS: Luis Guilherme da S. Costa, Andrei Camponogara, Wesley Mateus Cantarino, André Luiz Ferreira and Moisés V. Ribeiro
Dear Editor-in-chief and reviewers,
Thank you very much for your timely handling of our paper and the excellent review process. After revising the manuscript based on all the comments we have received, we would like to submit the revised manuscript entitled “A Notch Filter-based Coupling Circuit for Ultra Narrowband and Narrowband PLC
Applications” to be considered for publication in Industrial Sensors.
First of all, we kindly ask the editor for having the title changed to “A Notch Filter-based Coupling Circuit for Ultra Narrowband and Narrowband PLC Systems”.
In addition to the included changes for addressing all comments and suggestions posed by all the reviewers and the associate editor, we introduced additional modifications that may have increased our manuscript’s clarity and quality. The current manuscript includes several changes, and all changes have
been highlighted in blue.
We again appreciate the editor-in-chief and reviewers’ kindness in helping us improve the manuscript. All authors have reviewed and agreed to the submission of the revised manuscript. The changes made in the revised version of our manuscript are detailed in the point-by-point response to the Reviewers’
comments. Please do not hesitate to contact us if there are any other questions.
Below, please find our answers to each comment from the editor-in-chief and reviewers.
Yours sincerely,
The authors.
Response to Reviewer 1
• Comment 1: ”Please avoid the lumped references in the introduction section.”
We thank the reviewer for these helpful comments. We have rewritten several of the introduction section to eliminate this problem. Hopefully, the current version fulfills your expectations.
• Comment 2: ”The references are old. There are many recent papers related to the paper topic that must be added in the introduction section.”
We have been working on this subject for more than ten years and agree that several contributions have been reported in the literature in the last few years. However, a few of them show valuable content related to our investigation, at least from our perspective. Consequently, we prefer to include relevant references to advance this technology.
• Comment 3: ”The paper needs English improvements. Proofreading by a native English speaker should be conducted to improve both language and organization quality.”
We appreciate the reviewer’s comment, which helped us to revise the manuscript. Several parts of the manuscript were rewritten. Also, they were proofread by a native English writer.
• Comment 4: ”What are the research gaps in the literature? The authors did not identify the research gaps. The authors need to identify the research gaps, which they are planning to address in the paper.”
We appreciate the reviewer’s comment. We believe that the current version of the paragraph delimited by the lines 39 and 53 in the page 2/17 address this problem.
• Comment 5: ”The contributions of the paper must be better described. They must be benchmarked against the state-of-the-art to prove their novelty.”
We are thankful for your comment. First, we have rewritten the introduction section to clarify our contributions. Also, we rewrote Subsection 4.3 to correctly emphasize a comparison with an NB-PLC capacitive coupling circuit, a typical state-of-the-art coupling circuit for NB-PLC. Hopefully, the introduced changes fulfill your expectations.
• Comment 6: ”The conclusion should be revised. The reviewer strongly suggests improving the flow of the conclusion section. Start with a brief explanation of the paper, followed by a description of the main findings. Then finalize the conclusion with one or two suggestions for future work.”
Thanks for your review and suggestion. The conclusion section was completely rewritten. Hopefully, the introduced changes fulfill your expectations.
• Comment 7: ”The conclusion needs to be more efficient and contains the paper outputs and outcomes.”
Thanks for your review and suggestion. The conclusion section was completely rewritten. Hopefully, the introduced changes fulfill your expectations.

Reviewer 2 Report
1. The contributor needs to define the NB-PLC notch coupling circuit in detail.
2. Please provide the remaining section of this proposed article at the end of the introduction section for better understanding. The introduction section discussed various, but the requirements of the proposed model and how Notch Filter-based Coupling Circuit for UNB and NB PLC should be defined.
3. It Should have included the detailed elucidation of how to propos coupling circuit is more effective than a typical capacitive coupling circuit
4. The contributor needs to describe the functional description of the Twin-T notch analog filter in detail.
5. The contributor needs to provide enough information about which platform is used for the result analysis.
6. According to the manuscript, the author has made an excellent representation of addressing background issues, objectives, and methodology in the abstract section, which seems flawless for the readers.
7. In keywords, the given phrases look suitable, but they have to include some efficient terms.
8. Hence it will lead to a faultless review thoroughly. Similarly, in the introduction, the explanation about the exposure to UNB and NB PLC seems to be effective.
Author Response
RESPONSE TO REVIEWER FILE
Industrial Sensors
TITLE: A Notch Filter-based Coupling Circuit for Ultra Narrowband and Narrowband PLC Applications
MANUSCRIPT ID: SENSORS-1969792
AUTHORS: Luis Guilherme da S. Costa, Andrei Camponogara, Wesley Mateus Cantarino, André Luiz Ferreira and Moisés V. Ribeiro
Dear Editor-in-chief and reviewers,
Thank you very much for your timely handling of our paper and the excellent review process. After revising the manuscript based on all the comments we have received, we would like to submit the revised manuscript entitled “A Notch Filter-based Coupling Circuit for Ultra Narrowband and Narrowband PLC
Applications” to be considered for publication in Industrial Sensors.
First of all, we kindly ask the editor for having the title changed to “A Notch Filter-based Coupling Circuit for Ultra Narrowband and Narrowband PLC Systems”.
In addition to the included changes for addressing all comments and suggestions posed by all the reviewers and the associate editor, we introduced additional modifications that may have increased our manuscript’s clarity and quality. The current manuscript includes several changes, and all changes have
been highlighted in blue.
We again appreciate the editor-in-chief and reviewers’ kindness in helping us improve the manuscript. All authors have reviewed and agreed to the submission of the revised manuscript. The changes made in the revised version of our manuscript are detailed in the point-by-point response to the Reviewers’
comments. Please do not hesitate to contact us if there are any other questions.
Below, please find our answers to each comment from the editor-in-chief and reviewers.
Yours sincerely,
The authors.
Response to Reviewer 2
• Comment 1: “The contributor needs to define the NB-PLC notch coupling circuit in detail“
We thank the reviewer for their helpful comment. Improvements were introduced in Sections 1, 2, and 3 to address your concerns. Hopefully, the changes fulfill your concerns.
• Comment 2: “Please provide the remaining section of this proposed article at the end of the introduction section for better understanding. The introduction section discussed various, but the requirements of the proposed model and how Notch Filter-based Coupling Circuit for UNB and NB PLC should be defined.”
Special thanks for this comment. We have rewritten Sections 1, 2, and 3 to address this comment. The current writing clarifies the motivations, the problem, the proposal, and, most importantly, how the proposal addresses the presented problem. We hope your concern is addressed in our manuscript’s current version.
• Comment 3: “It Should have included the detailed elucidation of how to propos coupling circuit is more effective than a typical capacitive coupling circuit.“
We thank the reviewer for their thoughtful comment and effort toward improving our manuscript. In the current version of our manuscript, your question is very well addressed by the discussions related to Fig. 1 and Fig. 6. Actually, the discussion related to Fig. 1 fundamentally motivated the proposition of the NB-PLC notch coupling circuit. Hopefully, the new writings make it clear.
• Comment 4: “The contributor needs to describe the functional description of the Twin-T notch analog filter in detail.“
We sincerely appreciate this comment. To address it, we rewrote Subsection 3.3 to detail the Twin-T notch analog filter in Subsection 3.3 (i.e., formulation and functional description). We believe that additional details can be found in ref [41] Nilsson, J.W.; Riedel, S.A. Electric Circuits, 10 ed. pp. 586–587.
• Comment 5: “The contributor needs to provide enough information about which platform is used for the result analysis.“
Thank you for this suggestion. It would have been interesting to explore this aspect. Unfortunately, we have not used any platform to analyze our results. However, to address your concerns, we clarified that the design and simulation were carried out using the ADS software. Moreover, we show that the experiments were carried out in a lab facility in which equipment, such as the arbitrary function generator FY8300S [ref: 48] and the oscilloscope RTE 1204 Rhode & Schwarz [ref: 49] were used, see 2nd paragraph of Section 5. Hopefully, the improvements in Sections 3, 4, and 5 fulfill your expectations.
• Comment 6: “According to the manuscript, the author has made an excellent representation of addressing background issues, objectives, and methodology in the abstract section, which seems flawless for the readers.“
We appreciate the time and effort that you have dedicated to providing your valuable feedback on the manuscript. We are grateful for their insightful comments on paper.
• Comment 7: “In keywords, the given phrases look suitable, but they have to include some efficient terms.“
Thanks for this comment. We have rewritten the keywords. They are much better.
• Comment 8: “Hence it will lead to a faultless review thoroughly. Similarly, in the introduction, the explanation about the exposure to UNB and NB PLC seems to be effective.“
Thanks for your comments. Also, we have improved the quality of the presentation. Hopefully, the reviewer will see value in the introduced changes.

Reviewer 3 Report
Q1- Does the author need to explain analog notch filtering-based coupling circuits in more dentils to overcome the channel noise?
Q2- Does the author need to explain the search method used to reduce the computational complexity?
Q3- In the problem formulation section, the author should to explains in the subsection: 2.1 Channel Modeling 2.2 Noise Modeling
Q4- Should the author give more justification about increased attenuations Vs frequency with different (1 µF up to 11.16 µF)?
Q5-In Figures 3 and 4: the authors should give the following justification: a)Why the attenuations were dropped to minimum values versus frequencies? b)Should the author change the frequency of sales in fig4 such as fig3?
Q6- The author should justify why the attenuations decrease with different frequencies?

Author Response
RESPONSE TO REVIEWER FILE
Industrial Sensors
TITLE: A Notch Filter-based Coupling Circuit for Ultra Narrowband and Narrowband PLC Applications
MANUSCRIPT ID: SENSORS-1969792
AUTHORS: Luis Guilherme da S. Costa, Andrei Camponogara, Wesley Mateus Cantarino, André Luiz Ferreira and Moisés V. Ribeiro
Dear Editor-in-chief and reviewers,
Thank you very much for your timely handling of our paper and the excellent review process. After revising the manuscript based on all the comments we have received, we would like to submit the revised manuscript entitled “A Notch Filter-based Coupling Circuit for Ultra Narrowband and Narrowband PLC
Applications” to be considered for publication in Industrial Sensors.
First of all, we kindly ask the editor for having the title changed to “A Notch Filter-based Coupling Circuit for Ultra Narrowband and Narrowband PLC Systems”.
In addition to the included changes for addressing all comments and suggestions posed by all the reviewers and the associate editor, we introduced additional modifications that may have increased our manuscript’s clarity and quality. The current manuscript includes several changes, and all changes have
been highlighted in blue.
We again appreciate the editor-in-chief and reviewers’ kindness in helping us improve the manuscript. All authors have reviewed and agreed to the submission of the revised manuscript. The changes made in the revised version of our manuscript are detailed in the point-by-point response to the Reviewers’
comments. Please do not hesitate to contact us if there are any other questions.
Below, please find our answers to each comment from the editor-in-chief and reviewers.
Yours sincerely,
The authors.
Response to Reviewer 3
• Comment 1: “Does the author need to explain analog notch filtering-based coupling circuits in more dentils to overcome the channel noise?”
Thank you for this comment. If the spectrum of a high power additive noise occupies the frequency between 0 and fc, then the best solution is to assume that the channel is with erasure. On the other hand, If the high power spectrum content of the additive noise is beyond the fc, then the elliptic low-pass analog filter will reduce its influence. Note that this discussion was not included in the manuscript because it is more related to a data acquisition system-based contribution, which is beyond the scope of our investigation. Hopefully, the reviewer will understand our point of view.
• Comment 2: “ Does the author need to explain the search method used to reduce the computational complexity?”
Thank you for this comment. However, a discussion about computational complexity may not make sense because this paper is about electronic-based coupling circuits for receivers applied to UNB and NB PLC systems. Hopefully, the reviewer will understand our point of view.
• Comment 3: “ In the problem formulation section, the author should to explains in the subsection:
2.1 Channel Modeling 2.2 Noise Modeling.”
Thank you for this comment. Honestly, a discussion about channel and noise modelings is not appropriate for researching coupling circuits for PLC systems. Hopefully, the reviewer will understand our point of view.
• Comment 4: “Should the author give more justification about increased attenuations Vs frequency with different (1 µF up to 11.16 µF)?”
We agree with this and have incorporated your suggestion throughout the manuscript. The paragraph related to Fig. 1 was improved to clarify this issue (see paragraph between the lines 137 and 145). Hopefully, the current version fulfills your expectations.”
• Comment 5: In Figures 3 and 4: the authors should give the following justification: a)Why the attenuations were dropped to minimum values versus frequencies? b)Should the author change the frequency of sales in fig4 such as fig3?.
We found this comment difficult to understand. Anyway, we hope the improvements introduced in Section 4 cover what the reviewer has in mind.
• Comment 6: The author should justify why the attenuations decrease with different frequencies?
Thanks for your comment. The justification is that the NB-PLC notch coupling circuit operates as a very-narrow stop-band analog filter, in which the center frequency is the mains frequency. Therefore, this kind of information does not need to be presented because everyone with knowledge about analog or digital filters can quickly figure it out. Hopefully, the reviewer will understand our point of view.

Round 2
Reviewer 1 Report
The authors have considered all my concerns, hence the paper is accepted with its present form
Reviewer 3 Report
i accept paper in this form